# FedJETs: Efficient Just-In-Time Personalization with Federated Mixture of Experts

## Abstract

One of the goals in Federated Learning (FL) is to create personalized models that can adapt to the context of each participating client, while utilizing knowledge from a shared global model. Yet, often, personalization requires a fine-tuning step using clients' labeled data in order to achieve good performance. This may not be feasible in scenarios where incoming clients are fresh and/or have privacy concerns. It, then, remains open how one can achieve just-in-time personalization in these scenarios. We propose `FedJETs`, a novel solution by using a Mixture-of-Experts (MoE) framework within a FL setup. Our method leverages the diversity of the clients to train specialized experts on different subsets of classes, and a gating function to route the input to the most relevant expert(s). Our gating function harnesses the knowledge of a pretrained model (*common expert*) to enhance its routing decisions on-the-fly. As a highlight, our approach can improve accuracy up to 18% in state of the art FL settings, while maintaining competitive zero-shot performance. In practice, our method can handle non-homogeneous data distributions, scale more efficiently, and improve the state-of-the-art performance on common FL benchmarks.

## 1 Introduction

Due to the success of large-scale deep learning (27; 3; 4; 14; 5; 17), it is now widely accepted as a design philosophy that "*the larger (model/dataset), the better*". Yet, the increase of computational and memory costs that come from training such large models raises a key question: "*Are we spending the available budget wisely when we focus on training a single, monolithic model?*" Research stemming from the so called "grandmother cell hypothesis" in neuroscience (2) suggests that, ideally, a model's parameters should be specialized on different data (e.g., different features/classes/domains), such that we fully utilize the capacity of all parameters. This would potentially enable us to activate only part of the model on different data, resulting in sparse model activations and lower computation/memory/communication costs during training and testing (29; 5; 17; 26; 24; 31).

Mixture of experts (or MoEs)[1] (10; 11) is a famous sparse expert model variant that is motivated by the above premises. MoEs often utilize a gating function to activate parts of the global model, based on the current input data. Successful instances of MoEs include the works of (29; 5; 17; 26; 24; 31), where "experts" are defined as integral subparts of a neural network architecture (e.g., through a vertical decomposition of the fully-connected layers).

Yet, such a use of experts is not clearly motivated by data distribution learning, but rather as a means to scale up models (29; 5; 17; 26; 24; 31). This perspective is strengthened by recent advances on MoE research (34) that show, over a single domain dataset, a *random* gating function –which essentially removes any specialization of the experts– lead to trained models that perform favorably compared to more sophisticated gating functions. Works that aim to learn different data distributions through MoEs mostly focus on multi-domain tasks (18; 6), which require that *i*) we clearly distinguish between different data domains; and *ii*) each domain is an independent dataset for a separate expert.

---

[1]We distinguish MoEs from ensemble models (7): for the latter, "experts" are independently trained end-to-end, before models make decisions in an aggregated manner. In contrast, MoEs –either as submodels or as collections of disjoint models– are updated jointly before the testing phase.

Thus, *it is still an open question on how to put flesh on the promise of specialized experts for strictly better performance over a single domain dataset.*

Here, we study a new perspective to this quarrel, using Federated Learning (FL) as our workhorse. FL (22; 20; 12) is a distributed protocol that aims to achieve simultaneously three goals: *i*) to train a good global model that can generalize well across clients; *ii*) to enhance the global model's performance via good personalized models that can adapt to clients, where *iii*) these models should also work well on new incoming clients, ideally, without requiring additional data from them. Achieving these goals is not trivial, especially when the data distribution across clients is non-i.i.d. E.g., goal *iii*) could be achieved via fine-tuning, but this step requires access to *labeled* data, which may not be available, due to concerns on sensitive data. Alternatively, one could just use the global model as a *just-in-time/zero-shot model* for each testing client, which may not capture key aspects of incoming client's data.

**Main hypothesis and our contributions.** Applying MoEs in a single domain is meaningful –beyond scalability– if each expert can learn different aspects within the dataset. Our observation is that the non-i.i.d. data on each client can be viewed as having *different data characteristics*, on which different experts specialize. But, *how can we encourage the experts to discover such a specialization in a non-adhoc way?*

We propose `FedJETs`, a distributed system that connects and extends MoEs in a FL setting (See Figure 1). Our system is comprised of multiple independent models that each operate as *experts*, and uses a pretrained *common expert* model as feature extractor, that influences which experts are chosen for each client on the fly. These "ingredients" are combined with a novel gating functionality that guides the training: based on the common global model and current experts' specialization, the gating function orchestrates the dispatch of specific experts to be trained per active client, based on the local data. We argue that this approach can turn the bane of non-i.i.d. data into a blessing for expert specialization, as the experts can learn from diverse and complementary data sources and adapt to different client needs. Some of our findings include:

- `FedJETs` can exploit the characteristics of each client's local dataset and adaptively select a subset of experts that match those characteristics during training.
- `FedJETs` are able to dynamically select experts on-the-fly and achieve just-in-time personalization on unseen clients during testing. `FedJETs` accurately classify unseen data, not included in training, with small adaptations.
- We achieve these without violating data privacy, and by reducing the overall communication cost by not sending the whole MoE module to all clients, compared to state of the art methods (28).
- Some highlights of `FedJETs` in practice: `FedJETs` achieve $\sim 95\%$ accuracy on FL CIFAR10 and $\sim 78\%$ accuracy on FL CIFAR100 as a *just-in-time personalization* method on unseen clients, where the second best SOTA method achieves $\sim 71\%$ and $\sim 74\%$, respectively.

## 2 BACKGROUND

**Notation.** Vectors and matrices are represented with bold font (e.g., $\mathbf{x}$), while scalars are represented by plain font (e.g., $x$). We use capital letters to distinguish matrices from vectors (e.g., $\mathbf{W}$ vs $\mathbf{w}$). We use calligraphic uppercase letters to denote sets (e.g., $\mathcal{D}$); the cardinality of a set $\mathcal{D}$ is denoted as $|\mathcal{D}|$. Given two sets $\mathcal{S}_1$ and $\mathcal{S}_2$ that contain data, $\mathcal{S}_1 \neq_d \mathcal{S}_2$ indicates that the data in $\mathcal{S}_1$ does not follow the same distribution as that of $\mathcal{S}_2$, and vice versa. $[N]$ is $[N] = \{1 \dots N\}$.

**FL formulation.** Let $S$ be the total number of training clients. Each client $s$ has its own local data, denoted as $\mathcal{D}_s$, such that $\mathcal{D}_s \neq_d \mathcal{D}_{s'}, \forall s \neq s'$. We will assume that $\mathcal{D}_s = \{\mathbf{x}_i, y_i\}_{i=1}^{|\mathcal{D}_s|}$, where $\mathbf{x}_i$ is the $i$-th input sample and $y_i$ its corresponding label in a supervised setting. Let $\mathbf{W}$ denote abstractly the collection of trainable model parameters. The goal in FL is to find values for $\mathbf{W}$ that achieve good accuracy on all data $\mathcal{D} = \cup_s \mathcal{D}_s$, by minimizing the following optimization objective:

$$\mathbf{W}^\star \in \arg\min_{\mathbf{W}} \left\{ \mathcal{L}(\mathbf{W}) := \tfrac{1}{S} \sum_{s=1}^{S} \ell\left(\mathbf{W}, \mathcal{D}_s\right) \right\},$$

where $\ell\left(\mathbf{W}, \mathcal{D}_s\right) = \frac{1}{|\mathcal{D}_s|} \sum_{\{\mathbf{x}_i, y_i\} \in \mathcal{D}_s} \ell\left(\mathbf{W}, \{\mathbf{x}_i, y_i\}\right)$. Here, with a slight abuse of notation, $\ell\left(\mathbf{W}, \mathcal{D}_s\right)$ denotes the *local* loss function for user $s$, associated with a local model $\mathbf{W}_s$ (not indicated above), that gets aggregated with the models of other users. The local model $\mathbf{W}_s$ denotes a temporary image of the global model that gets updated locally at each client site, before sent to the server for aggregation. E.g., $\mathbf{W}_s$ could be a full copy of the global model at the current training round, or a selected submodel out of the global one, randomly chosen or based on client's characteristics.

It is desired that the trained global model $\widehat{\mathbf{W}} \approx \mathbf{W}^\star$ is applied to unseen test clients, that come with different non-i.i.d local data. Let $\mathcal{D}_{s'}$ denote the local data of a new client; then, $\mathcal{D}_{s'} \neq_d \mathcal{D}_s$ for all clients $s$ during training. Previous approaches handling a similar scenario (32; 30) assume we have access to part of the new client's labeled local data and fine-tune $\widehat{\mathbf{W}}$. We consider this as a limitation, since new users are likely unwilling/not able to provide accurate labeled data, and/or might not have sufficient resources to contribute to a fine-tuning phase of the whole model.

**The learning scenario we consider.** To be compatible with existing FL settings, we focus on image classification supervised learning tasks, as the most prevalent in literature. As a benchmark dataset, we use the CIFAR data suite (15; 8); following existing works, we partition the data samples by classes to turn full datasets into non-i.i.d. subsets. We assume the FL server-client protocol, where clients participate in training using local data; we assume there are 100 clients while we can only activate 10% clients per round. Our system deviates from traditional FL implementations (22; 20; 12); in those, one assumes a sole global model that is being shared with active clients, and updates to this model are being aggregated by the server per synchronization round. E.g., a large, ResNet-type of a network –like ResNet34, ResNet101 or ResNet200– could be used (9) in those scenarios. For our system, the "global" model is comprised by different independent ResNet34 models (9) that operate as experts, as well as a common pretrained ResNet34 model, that influences expert choice on the fly, and a novel gating function that guides both training and testing. In our scenarios, we assume a range of experts between 5 to 10. More details about our system in the section that follows.

## 3 Overview of FedJETs

**Overview.** Our system is depicted in Figure 1. On the server side (using purple boxes), we have access to a collection of experts (MoE module); see part **(a)** in Figure 1. These experts are selected to be of the same architecture (here, ResNet34, motivated by FL classification tasks) and their parameters are denoted as $\mathbf{W}_i$, $i \in [M]$. Further, these experts could be randomly initialized or could be a priori trained.

Beyond the MoE module, the server is responsible for a gating function; see part **(b)** in Figure 1. As we explain below in more detail, the gating function is a simple MLP defined by a set of parameters, denoted as $\mathbf{W}_r$.

On the client side (using cyan boxes), we assume that each client has access to the same pretrained expert; see part **(c)** in Figure 1. This *common expert* is used to embed local data to be further fed into the server's gating function; see part **(d)** in Figure 1. *Note that the common expert is never retrained during our procedure, but only used as an embedding mechanism.*

The result of the gating function is a sparse selection of experts; see part **(e)** in Figure 1. The selected experts, say experts $i$ and $j$, are communicated to the client (see part **(f)** in Figure 1), to be locally trained on the client side, where the selected experts' parameters –denoted as $\mathbf{W}_{i \in \mathbf{e}_s}$, with a slight abuse of notation– and the gating function's parameters, $\mathbf{W}_r$, are jointly trained; see part **(g)** in Figure 1. Finally, the updated parameters $\mathbf{W}_{i \in \mathbf{e}_s}$ and $\mathbf{W}_r$ are sent to the server to be aggregated with similar updates coming from other clients contributing in the same training round; see **(h)** in Figure 1.

At the core of our system lies the gating function that meets the following requirements:

- **Aim #1**: it should somehow "analyze" the data characteristics of local datasets;

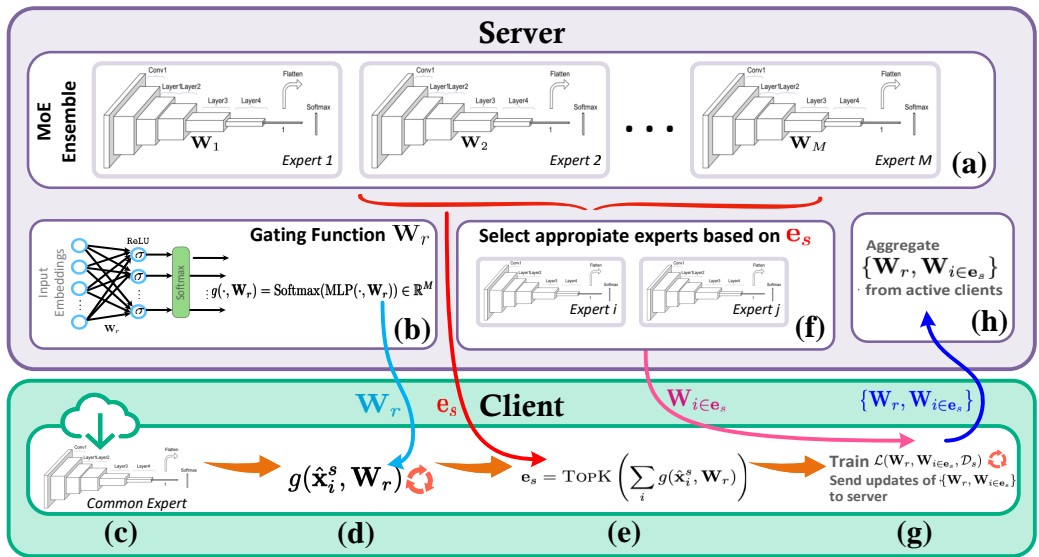

Figure 1: For each client: $i$) The server uses the gating function to select a subset of experts based on the local data distribution (parts **(a)**, **(b)**, **(d)**, **(e)**); $ii$) The client updates expert and gating function's weights (part **(g)**) and sends these back to the server. $iii$) the server aggregates and update the new weights (part **(h)**). The above are repeated for all FL rounds.

- **Aim #2**: it should guide the specialization of each expert;
- **Aim #3**: it should learn to fulfill the above, without sending all experts to clients during training.

**Aim #1**: *A new gating function.* Our gating function involves two parts: $i$) the pretrained common model that serves as a *feature extractor* from each client data samples. The embeddings are fed into the expert-ranking network. By design, our gating function should be model agnostic with respect to the pretrained common expert; such expert is considered a "black-box"[2], which does not need to be fully trained. $ii$) An *expert-ranking network* that predicts the specialization score of all experts for each data sample in local datasets. Such network is updated by each client, based on local data and a chosen subset of experts; the choice of experts is made before the local training starts. This function operates as a routing module that aggregates the score vectors from all the data samples in local datasets; per round, the routing module is designed to only select the most relevant experts to be sent to each active client per round.

**Aim #2**: *Guiding the specialization of experts.* We utilize a collection of experts that get updated locally by active clients. The selection of experts is made locally per client, based on the local data distribution, a common –across clients– expert and the current status of a global gating function. To achieve this second goal, we introduce a novel client activation strategy, called *anchor clients*, that $i$) encourages experts to have a meaningful initial specialization; and $ii$) enables the gating function to have a better understanding of each expert's specialization. To be more concrete, assume there are $M$ experts in our system; we also pre-select $M$ of clients (out of $N \ll M$) as *anchor clients*. Anchor clients are destined to be activated more often than normal clients, in order to form a specialized, one-to-one assignment relationship with experts, by using an *independent loss*. As we see in Section 5 this strategy stabilizes our system's performance.

**Aim #3**: *Efficient model training.* The idea of MoEs in FL is not new (28). Recent research (34) has demonstrated that directly using MoE methods (with or without top $K$ expert selection) yields similar performance to using a random gating function, which essentially eliminates any specialization of the experts. However, by leveraging the non-iid data in each client along with anchor clients to guide the training of the gating function and experts, we

---

[2]We never update the common expert, as it may be too large to distribute per round.

are able to overcome such random gating behavior and achieve true expert specialization. To the best of our knowledge, this is one of the first works to do so.

## 4 SYSTEM DETAILS

We dive into the details of our method in the following subsections.

**Pretraining.** Each client utilizes a pretrained common expert with parameters $\mathbf{W}_c$; see Figure 2. The common expert should have –to some extent– knowledge over the global data distribution $\mathcal{D} = \cup_s \mathcal{D}_s$. E.g., such an expert could be a pretrained model on CIFAR/ImageNet for image classification purposes. We restrict our methodology such that: $i$) we ensure our algorithm is agnostic to both the common expert's architecture and performance; $ii$) we assume

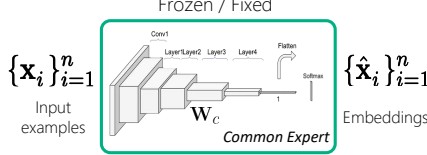

Figure 2: Common expert module.

only access to the common expert's embedding capabilities; and, $iii$) we do not modify/retrain the common expert. *The common expert is sent to all clients only once, before training.* For client $s$, we perform one-time inference on all local data using the common expert and store the corresponding output features for each data sample; noted as $\hat{\mathbf{x}}_i^s$ for each $\mathbf{x}_i \in \mathcal{D}_s$.

**The set of expert models.** Our methodology involves $M$ experts, each being an independent model of the same architecture.[3] For the $i$-th expert, $i \in [M]$, we denote its parameters as $\mathbf{W}_i$ and the corresponding model function as $f(\cdot, \mathbf{W}_i)$. See also Figure 1(a). The $M$ experts are randomly initialized in our experiments to provide full plasticity during training.[4] Each round, different subsets of experts are selected to be communicated to and updated by active clients, based on their local data (see Figure 1(f)). Per round, the updated experts are sent back to the server to be aggregated, before the next round starts; see Figure 1(h).

**The gating function.** We randomly initialize an expert-ranking network with parameters $\mathbf{W}_r$. This is a small-scale, two-layer MLP network that, per active client, takes the embeddings from the common expert based on local data, and predicts the specialization score of all experts. In particular, for client $s$ and given $M$ experts, we denote the score as $g(\hat{\mathbf{x}}_i^s, \mathbf{W}_r) \in \mathbb{R}^M$, for the $i$-th data sample, based on:

$$g(\hat{\mathbf{x}}_i^s, \mathbf{W}_r) = \texttt{Softmax}(\texttt{MLP}(\hat{\mathbf{x}}_i^s, \mathbf{W}_r)) \in \mathbb{R}^M.$$

The final decision on the top-$k$ experts is made via the rule:

$$\mathbf{e}_s = \text{TOPK}\Big(\sum_i g(\hat{\mathbf{x}}_i^s, \mathbf{W}_r)\Big), \quad \text{over all embedded local data samples, } \hat{\mathbf{x}}_i^s, \ i \in [|\mathcal{D}_s|],$$

where the TOPK$(\cdot)$ function selects the dominating experts, based on the current state of $\mathbf{W}_r$ and the local data embeddings $\{\hat{\mathbf{x}}_i^s\}_{i=1}^n$.

**The "anchor clients" mechanism.** Given $M$ experts, we pre-select $M$ special clients, with roughly distinct local data distributions, as *anchor clients*; we call all other clients as *normal clients*. For each anchor client[5], we initially pre-assign a one-to-one relationship to an expert; we denote the index of expert assigned to the $q$-th anchor client as $\mathbb{I}_q$ as an indicator function. At the beginning of each round, we follow the FL process, where only a small subset of clients, say $N$, is active; i.e., $N \ll S$, where $S$ is the total number of clients. Based on the discussion above, the active clients $N$ are split into $N_a$ anchor and $N_c$ normal clients, such that $N = N_a + N_c$. $N_a$ clients are activated from the pile of $M$ anchor clients, and $N_c$ clients from $(S - M)$ normal clients.[6] The idea is that, since $M \ll (S - M)$, we more frequently sample the anchor clients, which –combined with special expert assignment– shall encourage expert specialization. I.e., we encourage experts to be trained over the same data distributions of anchor clients to help specialization.

---

[3]This choice is made for simplicity. We consider diverse architectures per expert as future work.

[4]Discussion about different initialization for experts is provided in Section 5.

[5]Selection process for *anchor clients* is detailed on Appendix A

[6]Discussion about the ratio $N_a : N_c$ is provided in the experimental section.

**The training process.** To both normal and anchor clients, the server sends the current copy of parameters of the gating function, $\mathbf{W}_r$. The gating function then selects experts; the output $\mathbf{e}_s$ abstractly contains the set of chosen experts $\mathbf{W}_i$ for expert $s$, where $i \in \mathbf{e}_s$. The server receives $\mathbf{e}_s$ and sends the parameters $\mathbf{W}_i$, for $i \in \mathbf{e}_s$, to the corresponding client $s$; this routine reduces the communication cost –as compared to existing methods (28)– and encourages expert specialization.

Per training round, each normal client, using the standard cross entropy loss, will locally update both $\mathbf{W}_r$ and $\mathbf{W}_i$'s; the same procedure is followed for all active clients, each containing a different set of chosen experts. Formally, this amounts to (see also Figure 1, part **(g)**):

$$\mathcal{L}\left(\mathbf{W}_r, \mathbf{W}_{i \in \mathbf{e}_s}, \mathcal{D}_s\right) := \frac{1}{|\mathcal{D}_s|} \sum_{\{\mathbf{x}_j, y_j\} \in \mathcal{D}_s} \ell \left( \sum_{i \in \mathbf{e}_s} [g(\hat{\mathbf{x}}_j^s, \mathbf{W}_r)]_i \cdot f(\mathbf{x}_j, \mathbf{W}_i), \; y_j \right).$$

For an anchor client $q$, we only send the $\mathbb{I}_q$ expert to encourage expert specialization. Such an expert is trained regularly on the anchor's local distribution. Accordingly, we encourage the expert ranker network to recognize such rough specialization of the selected expert by using a simple independent loss. The two loss functions for anchor clients are as below:

$$\mathcal{L}\left(\mathbf{W}_{\mathbb{I}_q}, \mathcal{D}_q\right) = \frac{1}{|\mathcal{D}_q|} \sum_{\{\mathbf{x}_i, y_i\} \in \mathcal{D}_q} \ell\left(f(\mathbf{x}_i, \mathbf{W}_{\mathbb{I}_q}), y_i\right), \qquad \mathcal{L}\left(\mathbf{W}_r, \mathcal{D}_q\right) = \frac{1}{|\mathcal{D}_q|} \sum_{\{\mathbf{x}_i, y_i\} \in \mathcal{D}_q} \ell\left(g(\hat{\mathbf{x}}_i, \mathbf{W}_r), \mathbb{1}_{\mathbb{I}_q}\right),$$

where $\mathbb{1}_{\mathbb{I}_q}$ is the one-hot encoding indicating $\mathbb{I}_q$.

---

**Algorithm 1** FEDJETS

---

**Parameters**: $T$ rounds, $S$ training clients, $U$ testing clients, $M$ experts, $\ell_1$ local iterations, experts' function and parameters $f(\cdot, \mathbf{W}_i)$, gating function's function and parameters $g(\cdot, \mathbf{W}_r)$, common expert's parameters $\mathbf{W}_c$.

---

♠ **Pretraining** ♠
Send $\mathbf{W}_c$ to all clients;
// Data embedding
**for** $s = 1, \dots, S$ **do**
   $\hat{\mathbf{x}}_i^s = f(\mathbf{x}_i^s, \mathbf{W}_c)$
**end for**

---

♠ **Training** ♠
**for** $t = 0, \dots, T - 1$ **do**
   Activate $N_a$ anchor and $N_c$ normal clients;
   Send $g(\cdot, \mathbf{W}_r)$ to all activated clients;
   **for** $q = 1, \dots, N_a$ **do**
      Send expert $\mathbf{W}_{\mathbb{I}_q}$ to client $q$;
      **for** $l = 1, \dots, \ell_1$ **do**
         $\mathbf{W}_r^q = \mathbf{W}_r^q - \eta \frac{\partial \mathcal{L}(\mathbf{W}_r, \mathcal{D}_q)}{\mathbf{W}_r^q}$;
         $\mathbf{W}_{\mathbb{I}_q} = \mathbf{W}_{\mathbb{I}_q} - \eta \frac{\partial \mathcal{L}(\mathbf{W}_{\mathbb{I}_q}, \mathcal{D}_q)}{\mathbf{W}_{\mathbb{I}_q}}$;
      **end for**
-   **end for**

**for** $s = 1, \dots, N_c$ **do**
   Select a subset of experts $\mathbf{e}_s$ for client $s$;
   $\mathbf{e}_s = \text{TOPK}(\sum_{i=1}^{|\mathcal{D}_s|} g(\hat{\mathbf{x}}_i^s, \mathbf{W}_r))$;
   Send experts $\mathbf{W}_i, i \in \mathbf{e}_s$ to client $s$;
   **for** $l = 1, \dots, \ell_1$ **do**
      $\mathbf{W}_r^s = \mathbf{W}_r^s - \eta \frac{\partial \mathcal{L}(\mathbf{W}_r, \mathbf{W}_{i \in \mathbf{e}_s}, \mathcal{D}_s)}{\mathbf{W}_r^s}$;
      $\mathbf{W}_{i \in \mathbf{e}_s} = \mathbf{W}_{i \in \mathbf{e}_s} - \eta \frac{\partial \mathcal{L}(\mathbf{W}_r, \mathbf{W}_{i \in \mathbf{e}_s}, \mathcal{D}_s)}{\mathbf{W}_{i \in \mathbf{e}_s}}$;
   **end for**
**end for**

// Send to server for aggregation
$\mathbf{W}_r = \text{Aggregate}(\mathbf{W}_r^q, \mathbf{W}_r^s), \; \forall q, s$;
$\mathbf{W}_i = \text{Aggregate}(\mathbf{W}_{i \in E_q}, \mathbf{W}_{i \in \mathbf{e}_s}), \; \forall q, s$;
**end for**

---

♠ **Testing** ♠
**for** $u = 1, \dots, U$ **do**
   Send $g(\cdot, \mathbf{W}_r)$ and common expert, $\mathbf{W}_c$;
   $\mathbf{e}_u = \text{TOPK}(\sum_{i=1}^{|\mathcal{D}_u|} g(f(\mathbf{x}_i^u, \mathbf{W}_c), \mathbf{W}_r))$;
   Send experts $\mathbf{W}_j, j \in \mathbf{e}_u$ to client $u$;
   // Perform inference
   $j' = \max_{j \in \mathbf{e}_u} [g(f(\mathbf{x}_i^u, \mathbf{W}_c), \mathbf{W}_r)]_j$;
   $\hat{y}_i^u = f(\mathbf{x}_i^u, \mathbf{W}_{j'})$
**end for**

---

After all clients finish the local training round, the server applies a simple aggregation step to average the updated copies of $\mathbf{W}_r$ and $\mathbf{W}_i$'s. The above *is not trivial adaptation of MoE loss to FL settings*. After the selection of experts locally, the gating function can only "see" $K \ll M$ experts, facing a significant challenge: If it chooses all incorrect experts, this will lead to a decrease in performance and de-specialization of the selected experts. Our system though shows that, even with these restrictions, the gating function can overcome this difficulty and achieve near-perfect expert selection based on local data characteristics.

**Testing procedure.** During testing, we assume that we are given unseen new users with unseen local data distributions. We only send $K$ experts to each test client and we cannot

get access to local test data labels to perform fine-tuning. We first send $\mathbf{W}_r$ to the test client and select the top-$K$ experts, according to aggregated expert ranking score. Then, *for each test sample*, instead of using the weighted average of the output of all selected experts, we use the output of the expert with the highest expert ranking score to fully utilize the specialization of the expert. I.e., both experts might be utilized for different data samples, instead of averaging their performance on each testing sample.

The above are summarized in pseudocode in Algorithm 1.

## 5    EXPERIMENTS

**Task and model description.** For the experts' architecture, we use ResNet34 (9). For the gating function, we use a two-layer MLP followed by a SoftMax layer at the output to weight each expert. For the clients, we use the SGDM optimizer, with learning rate 0.01 and momentum 0.9; we set the batch size to 256 and the number of local epochs to 1. For the gating function update, we use the SGD optimizer with learning rate 0.001. The aggregation of the model weights on the server side is performed with FedAvg (23).

**Dataset.** We conduct experiments on CIFAR10 and CIFAR100 (15). Initially, the training dataset is randomly partitioned across 100 clients. We followed the same procedure for the *anchor* clients but we avoided replacement, aiming to preserve the label diversity in each subset. We establish a one-to-one mapping between these clients and the experts, corresponding to each group of labels. This path leads to: *i*) we have one expert available for each group; and *ii*) we retain flexibility to activate the *anchor* clients during the training rounds. The complete client distribution for both datasets is detailed in the Appendix A.

**Zero-Shot Personalization.** Let us first describe the baselines to compare against:

- `FedMix` (28) trains an ensemble of specialized models that are adapted to sub-regions of the data space. By definition, `FedMix` sends all the experts to each client in order to specialize them, heavily increasing communication costs. For this implementation, we initialized the common expert from the initial pretrained model checkpoint and we use it to embed local data in the gate function and help the routing.

- `FedAvg`(23) is the de facto approach for FL and allows to have a fair comparison in terms of fixed communication cost. Here, we initialize the global model with the initial common expert checkpoint, and aggregate the updates from all sampled clients per iteration.

- `FedProx` (21) tackles heterogeneity by introducing a regularization term that limits the distance between local/global model, at the cost of additional computation overhead per round. For initialization, we follow the same strategy with `FedAvg`.

- `Scaffold` (13) handles non-iidness by applying control variates for the server and clients at the expense of doubling communication cost per round compared to `FedAvg`. This method tends to become unstable during training, as previous studies have shown (19). For initialization, we follow the same strategy with `FedAvg`.

- The `Average Ensembles` (16) train two models (initialized from the common expert) as in `FedAvg`, but with different random seeds. It then combines them by averaging outputs probabilities. While it provides flexibility w.r.t. resources, it has higher inference costs.

| | | CIFAR 10 | | | | | CIFAR 100- Running | | | | |
|---|---|---|---|---|---|---|---|---|---|---|---|
| Method | # Clients | Rounds | $M$ | $K$ | Acc. | Acc. | Rounds | $M$ | $K$ | Acc. | Acc. |
| Common Expert | – | – | – | – | 73% | 93% | – | – | – | 67% | 73% |
| FedMix (28) | 100 | 1250 | 2 | 2 | 31.3% | 42.9% | 2000 | 2 | 2 | 49.7% | 48.3% |
| FedAvg (23) | 100 | 1250 | – | – | 31.2% | 58.4% | 2000 | – | – | 72.9% | 74.0% |
| FedProx (21) | 100 | 1250 | – | – | 72.7% | 71.4% | 2000 | – | – | 72.8% | 74.0% |
| Average Ensembles (16) | 100 | 1250 | – | – | 23.9% | 53.7% | 2000 | – | – | 72.8% | 74.1% |
| FedJETs | 100 | 1250 | 5 | 2 | **91.8%** | **95.7%** | 2000 | 10 | 2 | **75.7%** | **78.6%** |

Table 1: Average zero-shot personalization score for unseen test clients. The results are presented using two different pretrained *common experts* as feature extractor for each dataset: a) The lower bound model at which the gating function is able to outperform the initial common expert accuracy, illustrated in Figure 4; b) The average model that represents a good accuracy that is relatively easy to achieve using ResNet-34 architecture. Sampling is performed under the scheme of $N_a = 5$ *anchor* + $N_c = 5$ normal clients per training round; here, $N = N_a + N_c = 10$. Appendix D contains a thorough assessment of the `FedJETs` gating function's effectiveness on individual samples.

Table 1 summarizes our findings on this setup. Whereas `FedMix` requires all experts be transmitted to each client, i.e., $M = K$, `FedJETs` allows the selection of $K$ experts, here $K = 2$, enabling a larger battery of experts without having to send them all. This not only reduces communication costs, but also ensures that the client is receiving the most pertinent information from the relevant experts.

In terms of baselines, we observe that both datasets behave differently. We attribute this gap to the number of classes each client holds. In the CIFAR10 scenario, each client has fewer classes, which can amplify the model drift problem in all baselines. Furthermore, `FedAvg`'s performance deteriorates sharply, when we test it on the new CIFAR10 clients that were not used for training, due to the heterogeneous data distribution during training and then in testing phase. Similarly, `Average Ensembles` faces a performance ceiling, as the ensembles inherit the limitations of the `FedAvg` aggregation method. On the other hand, `FedProx` is able to surpass the initial performance of the common expert for the CIFAR100 scenario, but degrades quickly when using few labels per client as in the CIFAR10 setup. To the best of our ability, we attempted multiple hyperparameter settings for `Scaffold`, yet we were unable to produce a useful model under this distribution; it became unstable during training (10% for CIFAR10 / <5% for CIFAR100). Further comparison against domain adaptation methods, as in `FedADG` (33) and `FedSR` (25), is shown in Appendix G; for the cases we consider, we observe that current implementations are bound to having a small number of clients in order to perform competitively.

The global accuracy reported at the end of training demonstrates the effectiveness and consistency of `FedJETs` in both datasets, with significantly better performance than other algorithms. Please refer to Appendix B for a detailed end-to-end performance of the methods in Table 1 under different clients' distribution.

**Ablation study: Initial common expert impact.** We conduct a thorough evaluation of the performance tradeoffs, when utilizing different common experts for the gating function decisions. Our findings indicate that the amount of training allocated in the initial *common expert* has a critical effect on the overall performance of `FedJETs`. E.g., if the gating function uses a poor common expert for training, it can lead to poor performance (collapses to selecting a single expert), and therefore not be able to improve beyond the baseline.

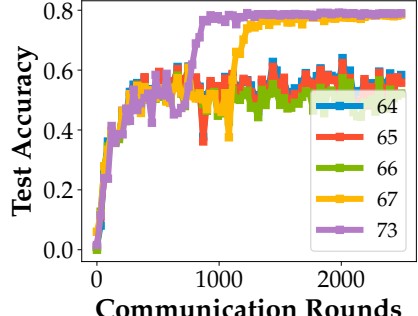

Figure 3: `FedJETs`'s performance on CIFAR100 dataset, using different initial accuracy for *common expert* (legends of the plot); the setup in Table 1 is used.

Figures 3-4 show that the breakpoint of the gating function for the CIFAR100 dataset is approximately 66% accuracy by the *common expert*. In Figure 4, it becomes clear that a major cause of this breakpoint is the fact that most of the experts are unable to surpass the initial accuracy of the common expert. This is attributed to the lack of an effective selection of experts, which is essential for the gradient updates of each expert to be aligned with the same part of the task. Figure 3 also reveals the following: the 67% case, given a few more iterations, is able to match the performance of the 73% case. This suggests a "phase-transition" might exist, where more effort (i.e., communication) is needed to improve beyond the common expert's performance. This implies also the performance of `FedJETs` depends on the quality of the experts.

**Ablation study: Common expert boosts experts' performance.** In order to test this hypothesis, we initialize each expert from the common expert and continue training for 2000 rounds. In Table 2, we observe the final score of each method. Surprisingly, for `FedJETs` it takes a few more rounds to overcome the baseline than when the experts are initialized from scratch. This is because the pretrained model is optimal to the entire dataset. In order to successfully specialize each expert, it is necessary to retrain the model on the specific subset of labels.

| Common Expert | 73.73% |
|---|---|
| FedMix | 73.78% |
| FedAvg | 73.99% |
| Average Ensembles | 74.10% |
| FedJETs | **83.27%** |

Table 2: Average zero-shot accuracy for CIFAR100 after 2000 rounds.

We also plot in Figure 4 the performance of each expert (denoted as expX) over the communication rounds for different initial accuracy of the common expert. It is obvious that, for our setting, using a common expert with an accuracy below 67% does not allow the gating function to improve sufficiently, thus preventing experts from improving beyond the baseline. Once the gating function can utilize a slightly better common expert, we are able to outperform the rest of the methods.

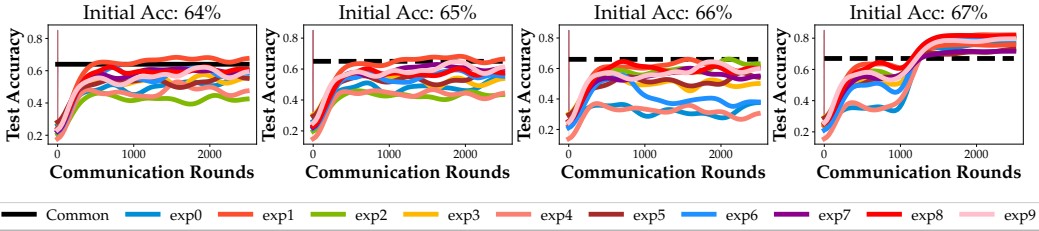

Figure 4: Zero-shot personalization accuracy per expert during training on CIFAR100.

**Ablation study: The "anchor/normal" client ratio.** To ensure the best performance of `FedJETs`, the sampling scheme must be carefully studied. This is due to the fact that each expert has a distinct distribution; i.e., their local objectives are only aligned with a particular subset of labels. It is essential to ensure consistency in the experts' updates to prevent them from drifting away from their own "task". As previously mentioned, we assume we have some control over the activation of the clients during training.

Our solution is the proportional introduction of the *anchor* clients, whose main purpose is to act as regularizers, ensuring consistency in the expert updates during training. To find the optimal ratio of anchor/normal clients $\frac{N_a}{N_c}$ we conduct experiments varying this ratio; see Figure 5. Sampling half of the clients per round as *anchor* quickly surpasses the baseline of the common expert and maintains high consistency in subsequent iterations. Using a lower ratio of 30% anchor clients per round also achieved similar performance, allowing some flexibility in the sampling. Contrarily, when we sampled clients randomly from the available pool (i.e., no "anchor clients"), `FedJETs` shows difficulty improving performance, as experts' updates become inconsistent. Appendix C shows the end-to-end performance difference across different methods using these sampling ratios for both datasets.

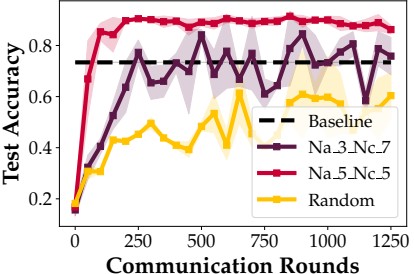

Figure 5: `FedJETs_Na_X_Nc_Y` means that $\frac{X}{Y} = \frac{N_a}{N_c}$, and $N = N_a + N_c$. 30% anchor/normal client ration is enough to match baseline accuracy, however the model becomes more inconsistent by converging slower.

## 6 CONCLUSIONS

`FedJETs` is a novel distributed system that leverages multiple independent models –in contrast to recent MoE applications, where parts of a single model are considered as experts– and a pretrained common expert model to achieve just-on-time personalization in applications with diverse data distributions, as in FL. Unlike existing methods that rely on predefined or fixed expert assignments, `FedJETs`, via a novel gating functionality, can dynamically select a subset of experts that best suits each client's local dataset during training. Experiments show that `FedJETs` achieve $\sim 95\%$ accuracy on FL CIFAR10 and $\sim 78\%$ accuracy on FL CIFAR100 as a *just-on-time personalization* method on unseen clients, where the second best state of the art method achieves $\sim 58\%$ and $\sim 74\%$, respectively.

Overall, our work contributes to the development of more efficient and flexible ML systems that, not only learn from, but also specialize on distributed data. Our approach can be extended to other domains and applications –such as natural language processing models– and we plan to investigate the potential of `FedJETs` in those areas as future work. Theoretically understanding the mechanism behind `FedJETs` in simple scenarios is also considered an interesting future research direction. We hope that our work will inspire further research on this path and help build a more sustainable and equitable AI ecosystem.

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
