# OpenReview forum: "FedJETs: Efficient Just-In-Time Personalization with Federated Mixture of Experts"
_ICLR.cc/2024/Conference — Submitted to ICLR 2024_

### Official Review · Reviewer_Na6T · 2023-10-26

**Soundness:** 2 fair
**Presentation:** 3 good
**Contribution:** 1 poor
**Rating:** 3
**Confidence:** 5

**Summary:**

The paper uses Mixture of Experts architecture with a gating function to select "the most relevant" experts for each client data "just-in-time" for federated learning. They also take advantage of a pretrained model as "common expert". The authors aim i) global generalization ii) enhance global model via personalized models ii) solve "cold-start" problem.

**Strengths:**

- It is good to see that the authors have used MoE for federated learning, differently from the FedMix paper.
- Reducing communication costs is critical and very good.
- Using anchor users seems useful.

**Weaknesses:**

- The contribution lacks novelty, as using a gating function and common expert is not new. Even there are multi-gate mixture of experts architectures in the literature [1].
- The architecture is similar to STAR model in paper [2] without anchor users.
- There are couple of places mention that our approach lower communication costs but there is no experimental results that show that how much improvement is there as in FedMix paper.
- There are no experiments for the cold-start problem as claimed in the paper ( this is not same as unseen new users for testing. Testing, of course, should be unseen).

[1] Modeling Task Relationships in Multi-task Learning with
Multi-gate Mixture-of-Experts (https://dl.acm.org/doi/pdf/10.1145/3219819.3220007)
[2] One Model to Serve All: Star Topology Adaptive Recommender
for Multi-Domain CTR Prediction (https://dl.acm.org/doi/pdf/10.1145/3459637.3481941?casa_token=X928_yKMvcsAAAAA:WKNfD3i-ELk5CTxjIqs8t6MxMN0LSmwwhIvbEY7lvKaoqp8BC0zQdUOuZHXQKUkMUH1poak8ZFxZ)

**Questions:**

- The claim "we partition the data samples by classes to turn full datasets into non-i.i.d. subsets", how do you make sure that samples with different class labels with same data is non - i.i.d ?
- This work also very similar to multi-task learning, one of the main problem is conflicting gradients. Since you claim the data is non i.i.d. have you ever encountered this problem as in these papers [3] [4]

[3] MAMDR: A Model Agnostic Learning Framework
for Multi-Domain Recommendation (https://dl.acm.org/doi/pdf/10.1145/3459637.3481941?casa_token=X928_yKMvcsAAAAA:WKNfD3i-ELk5CTxjIqs8t6MxMN0LSmwwhIvbEY7lvKaoqp8BC0zQdUOuZHXQKUkMUH1poak8ZFxZ)
[4] Gradient Surgery for Multi-Task Learning (https://proceedings.neurips.cc/paper/2020/file/3fe78a8acf5fda99de95303940a2420c-Paper.pdf)
[5] Conflict-Averse Gradient Descent
for Multi-task Learning (https://proceedings.neurips.cc/paper_files/paper/2021/file/9d27fdf2477ffbff837d73ef7ae23db9-Paper.pdf)

---

> ### Author Response · Authors · 2023-11-21
> **Response to reviewer Na6T**
>
> We want to thank the reviewer for the detailed comments. Below, we respond to the points raised one by one. We hope our responses resolve further concerns and are available for other questions.
>
> -**Regarding lack of novelty:** We understand your point of view; however, we believe our paper has a different focus than the references pointed out in the following ways:
>
>   -*MMoe* [1]: [1] proposes to learn task relationships by sharing ALL expert models across ALL tasks while using different gating functions that learn task similarities across all the experts. FedJET's main advantage is that it dynamically selects the most relevant experts (according to the client distribution) and only sends these models across the network, avoiding the communication costs of sending all the experts as FedMix. On the other hand, FedJETs uses a single gating function to learn the differences across different domains. At the same time, MMoe utilizes multiple gates (one per task) that learn how to model the different relationships across the experts and weigh them so they can be used differently.
>
>   -*Star Topology* [2]: [2] proposes a method that constantly updates two models during training: a) Global model, which shares parameters across all personalized models, and b) In-domain personalized models (1 per domain). In FedJETs, the "pretrained model" remains frozen at all times, it is never updated during training or testing, and its only purpose is to serve as a feature extractor and feed its embeddings to the gating function so that it can rank the expert models. During inference, [2] combines these weighted networks and unifies them into a new one via an element-wise multiplication process. FedJETs ranks the subset of matching experts per sample level, capitalizing on a single expert more relevant to the sample distribution. Lastly, in [2], the domain ID needs to be explicitly fed into the network to facilitate the model learning; FedJETs dynamically identify the relevant experts by training the gating function.
>
> -**Regarding lower communication cost**:
> In our experiments, we demonstrate how our method is more communication efficient than FedMix by showing how our approach can achieve significantly better final performance using the same communication budget shown in Tables 1 and 5 (in the appendix). In Table 5, if we compare FedJETs using two experts with FedMix using five experts, we can see FedJETs achieve better final performance with a smaller communication budget.
>
> -**Regarding cold-start problem**:
> Our paper does not use the term "cold-start" for our targeted scenario. Our method will adapt to new clients with unseen local dataset distributions and without labels during testing by dynamically selecting experts. We are not using any label from the test client, so we define this as zero-shot penalization.
>
> -**Regarding data non-iidness**:
> We reproduce two main partitioning strategies proposed in [19] to simulate the non-iidness in our clients. The first partitioning is quantity-based label imbalance, and the second uses a more standard approach, distribution-based, which simulates the Dirichlet function across the total number of clients. All the details about the clients' partition can be found in Appendix A, and a more detailed evaluation of FedJETs under these distributions is included in Appendix B.
>
> -**Regarding conflicting gradients**:
> During experiments, we discovered that anchor clients act as regularizers and help maintain consistency in the experts' updates when the gating function makes the wrong selection. The pretrained model - permanently frozen - serves as a feature extractor; these embeddings feed the gating function, ensuring that only the relevant experts are sent to each client and the expert updates are directed towards the same objectives, thus avoiding model drift. Additionally, there is an initial degree of randomness in the gating function. During the first couple of iterations, the gating function sends random top $K$ experts to each client while the experts learn to specialize in the different regions of the label space. We found a way to keep consistency during these initial rounds: through the anchor clients. By introducing at least 30\% anchor clients during each training round, we can ensure a balance between the wrong selection of the gating function by allowing them to act as regularizers in light of conflicting updates on the experts. Appendix C provides a detailed explanation of this phenomenon encountered during training.

---

> ### Comment · Reviewer_Na6T · 2023-11-23
>
> Thanks for your response. I will keep my score.

---

### Official Review · Reviewer_k6Nb · 2023-11-01

**Soundness:** 3 good
**Presentation:** 3 good
**Contribution:** 2 fair
**Rating:** 6
**Confidence:** 3

**Summary:**

The paper studies a federated learning setting where the goal is to fine-tune the models. The main framework FedJETs is given a pretrained model and contains multiple ``expert'' models and a gating function. When new client data comes in, the gating function utilizes the representation from the pre-trained model to decide which K experts to update. Then, using the client's data, FedJETs obtain updates for the gating function as well as the K experts and send them back to the server. The server aggregates and updates the new weights.

**Strengths:**

The paper presented the main idea as well as the FEDJETs algorithm in a clear and intuitive manner. The idea of having individual expert models and a gating function to select experts is intuitive and reasonable. The authors also discussed the technical difficulties coming with this design. The experimental results suggest the efficacy of the proposed method.

**Weaknesses:**

My biggest concern is the novelty of the proposed method. The general framework of having individualized models and selecting a subset of experts for performing ensemble learning is a traditional topic. The specific setting of having a pre-trained model along with a gating function to select a subset of experts to update is new. I am not entirely familiar with the current federated learning literature, so I will leave other reviewers to decide on the novelty of the paper to the federated learning community.
In addition to the concern about the novelty of the work, another concern I have is the applicability of the method when expert models need to be very large. It seems to be inefficient to use the common expert (a large pre-trained model) to just perform expert selection. Would it be more reasonable, computation-wise at least, to not have individual expert models but different expert heads so that the pretrained common expert can be used to extract a common representation to pass into different experts?

**Questions:**

- How should the number of experts scale with the number of clients? How should one choose the ``K'' hyperparameter?
- Could the authors comment on the computation and memory costs of having individual experts? How big should the expert model be compared to the pretrained common expert model?
- Other than using the pre-trained model for obtaining representation for the gating function, is it used in some other ways, e.g., is there a way to combine its output with the expert model?

---

> ### Author Response · Authors · 2023-11-21
> **Response to reviewer k6Nb**
>
> We want to thank the reviewer for the detailed comments. We address each point raised and hope our responses will alleviate further concerns. We remain available to answer any additional questions you may have.
>
> -**Regarding differences with ensemble learning**: There is a critical difference between ensemble learning and FedJETs: while the former combines multiple models to create a more accurate prediction during testing, FedJETS uses a weighted average per sample on all the selected $K$ experts, and utilizes the highest expert ranking score to fully capitalize on the specialization of a single expert at the sample level. Algorithm 1 (Testing) shows this is not a trivial adaptation of ensembles. Also, Appendix D provides further details on the gating function behavior during inference; if required, we can move some of this material into the main text to make ideas emerge more naturally. To the best of our knowledge, though, we are unaware of any works that learn mechanisms to select on-the-fly subsets of experts to be combined during inference; if there is such literature we are missing, we are eager to know.
>
> -**Regarding large-scale models, efficiency, and pretrained models**: First, we note that one of the main advantages of our method is that the ``pretrained common expert'' is considered a black box. This means that the architecture of the pretrained model is not strongly coupled with that on the expert's side. This allows us to leverage the knowledge of any pretrained model with capabilities over the different domains on the dataset. Further, this allows us to not rely on the model's full capacity; therefore, we assume this model is not extremely large, and neither has to be fully trained. Yet, it is noteworthy that during the early stages of our design, we experimented with the case where information (e.g., the whole model or part of the model, or even embeddings) from the pretrained model was available to be used along with the experts selected. Yet, we noticed that this often led to performance degradation, and the experts required further training to achieve some specialization.
>
> -**Regarding expert/clients ratio**: The experts in FedJETs are agnostic to the total number of clients. Experiments presented in Table 1 demonstrate the behavior of FedJETs for various $M$ values (total number of experts) with a fixed number of clients $K$. Each expert should have one "anchor" client that matches its specialization range to ensure that the gating function learns to assign clients to the right experts in the early stages of training. Appendix A, Figure 6, shows an example of how we randomly assigned specialization ranges to the experts. We hope this information clarifies the reviewer's concern.
>
> -**Regarding computational/memory costs**:
> The architecture of the experts and that of the pretrained model are not necessarily coupled; instead, the pretrained model operates as a black box that decides which experts should be selected and activated. Further, the pretrained model does not require full training, as it is used as a feature extractor. Conversely, since the experts specialize in specific labels of the input space, the model size can be considered "small" compared to a monolithic model that covers all labels.
>
> -**Regarding combining the expert model with that of pretrained model**:
> While we do not exclude the possibility that the pretrained model can be somehow combined with expert models, as we mentioned above, we have observed a performance degradation in our initial attempts. This is an exciting research direction that we will be closely looking into in the near future.

---

### Official Review · Reviewer_pX49 · 2023-11-01

**Soundness:** 2 fair
**Presentation:** 3 good
**Contribution:** 2 fair
**Rating:** 5
**Confidence:** 3

**Summary:**

The paper proposes FedJETs, a distributed system that connects and extends Mixture of Experts in FL setting. The system features multiple independent models as experts, in contrast to common MoE settings where different parts of a model is considered as experts. The authors introduce a pretrained common expert and a novel gating functionality to guide the specialization of experts during training. The authors claim that the combined system can exploit the characteristics of each client’s dataset and adaptively select experts suitable during training. FedJETs also claims to be able to dynamically select experts and adjust to unseen clients on-site.

**Strengths:**

An interesting combination of Federated Learning scenario and the idea of Mixture of Experts. Viewing independent models as separated experts and guiding them respectively during training rounds of an FL setup can serve as a new attempt, although similar ideas can be found in certain meta-learning scenarios. The presentation of methods and the clarity of expression are good.

**Weaknesses:**

Some of the experiments supporting the proposed method might not be considered as sufficient. For FL scenarios, there are plenty of available datasets beyond the CIFAR data suite with more obvious levels of Non-IID features (e.g., the LEAF benchmark datasets). More results on such datasets might be appreciated considering the nature of this paper. Besides, the ablation study regarding the anchor client ratio might not be sufficient as to determine the claimed “optimal” ratio. It served the propose to address the significance of anchor clients, but there could be more to explore regarding such a key component of the entire method.

Updated After Response

Thanks for your response! Only CIFAR and the added EMNIST are not sufficient, considering many other large-scale datasets have natural client partitions.

**Questions:**

It would be appreciated, considering the nature of this paper, if more results regarding Non-IID datasets other than the CIFAR data suite could be demonstrated. For FL scenarios, there are plenty of available datasets beyond the CIFAR data suite with more obvious levels of Non-IID features (e.g., the LEAF benchmark datasets).

Besides, the ablation study regarding the anchor client ratio might not be sufficient as to determine the claimed “optimal” ratio. It served the proposal to address the significance of anchor clients, but there seems to be more to explore regarding such a key component of the entire method. Is it possible for a higher anchor-normal client ratio to achieve faster convergence or even a better overall performance?

---

> ### Author Response · Authors · 2023-11-21
> **Response to reviewer pX49**
>
> We want to thank the reviewer for the detailed comments. Below, we respond to the points raised by the reviewer, one by one. We hope our responses will resolve any further concerns, and we are available for any other questions.
>
> -**Regarding more datasets**:
> First, we agree with the reviewer that including more datasets can help to illustrate the differences among different domains better. For this reason, we provide new results on the EMNIST dataset (table below) using the same setup as Table 1 in the paper, which further demonstrates the advantage of our method; note that completing the existing experiments in the paper took a significant amount of GPU time, given the excessive ablation study results we present. Moreover, we point out that the CIFAR data suite is still highly relevant for current benchmarks on Federated Learning. Table 1 shows a noticeable difference in behavior between the two datasets; CIFAR10 has more difficulty reaching the desired performance due to its fewer classes, exacerbating the model drift problem in all baselines. FedJETs is the only method able to surpass the initial accuracy of the Common expert, whereas the rest only degraded during training.
> | FedJETs | FedMix |
> |---------|--------|
> 0.9639 | 0.9259
>
> -**Regarding anchor clients ratio**:
> The ablation study examines the importance of the ``anchor clients" during sampling, i.e., the clients that share the same expert for the target task. It shows that we need at least 30\% of anchor clients to guarantee that our method can surpass the initial accuracy of the pretrained common expert. Still, Figure 5 also explores two other scenarios: $i)$ How does our approach perform if we randomly sample clients across the training iterations without any control over the expert distribution? And $ii)$ How does our method benefit from increasing the proportion of anchor clients to 50-50? While the latter ensures faster convergence of our method, it is not very realistic, as enforcing higher ratios of anchor clients in federated learning may affect the efficiency/deployment costs and privacy. Appendix B (Figure 9) shows the detailed performance of our method under $i$ and $ii)$ setups for both datasets.

---

### Official Review · Reviewer_Y7tu · 2023-11-30

**Soundness:** 2 fair
**Presentation:** 3 good
**Contribution:** 2 fair
**Rating:** 5
**Confidence:** 3

**Summary:**

The paper proposes a novel solution for efficient just-in-time personalization with Federated Mixture of Experts, called FedJETs. The method leverages the diversity of clients to train specialized experts on different subsets of classes, and a gating function to route the input to the most relevant expert. The approach can improve accuracy by up to 18%, while maintaining competitive zero-shot performance.

**Strengths:**

- The introduction of MoE in the field of federated learning has enabled neural networks to specialize in various types of datasets. Additionally, it achieves non-linear growth in testing time while increasing the model parameter scale.
- Without compromising data privacy, the overall communication cost is reduced by not transmitting the entire MoE module to all clients.
- The anchor clients mechanism addresses, to some extent, the inherent challenge in MoE where only a subset of experts receive sufficient training.

**Weaknesses:**

- The innovation is not sufficient to meet the criteria for publication at ICLR. Essentially, the approach merely applies MoE to federated learning, breaking down a single model into multiple based on different dataset features.
- The experimental section fails to elucidate the advantages of the anchor clients mechanism over the method of selecting fixed experts for specific datasets.
- The mentioned reduction in communication volume by not loading the entire MoE onto clients, as stated earlier, is not reflected in the experiments.

**Questions:**

1. Could the consideration of replacing the gate function with the anchor clients mechanism be explored and the impact of the gate function on experimental results analyzed? The previously mentioned non-i.i.d. datasets, representing datasets with different features that perform well in MoE, could be further validated through experiments with additional comparative trials involving random routing to assess MoE performance.

2. During the model training process, clients select expert networks and transfer them from the server to the client for distributed parameter training. After the parameters are trained, they are sent back to the server for synchronization. While this method ensures privacy, has the potential increase in communication latency been considered? Can this method's latency be experimentally compared and analyzed against traditional methods?

**Details Of Ethics Concerns:**

Nan

---

### Meta-Review · Area_Chair_CiiB · 2023-12-02

**Metareview:**

Reviewers had slightly diverging recommendations on this work, but a recurring concern (even amongst otherwise positive reviews) was the novelty of the proposal, being the use of a MoE model in a federated learning setting. Another concern was whether the empirical results fully reflect the advantages of expert selection versus usage of a fixed expert. The paper could benefit from a more detailed discussion contrasting to prior proposals, a clearer demonstration of the value in terms of communication cost, and additional results on large-scale datasets.

**Justification For Why Not Higher Score:**

No strong champions for the work. Recurring concerns around novelty.

**Justification For Why Not Lower Score:**

N/A

---

### Decision · Program_Chairs · 2024-01-16

Reject